# Peer review of "Essential renormalisation group"

_SciPost Physics_

## Round 3 · Referee Report · Anonymous · 2022-5-23

Report
The authors have done a substantial revision of their manuscript and they have taken into account many of the suggestions I formulated in my first report. Still, a few of my reservations remain:
One is related to my original point (4). I still find the "bootstrap" argument propagated here by the authors unsatisfactory and would believe that such transformations that can actually be done should be understandable fully and explicitly. With the implicit construction I see the danger that one assumes that certain transformations can be realized and that only a later more careful investigation will show the mistake.
Related to this I still believe that the suggestion in my original point (5) would be the better way to proceed. Once the quantum effective action (or effective average action) is defined, linear or non-linear field redefinitions (also k-dependent) are possible (with the restriction that convexity might be affected by non-linear transformations). This is actually also the kind of transformations discussed by Weinberg in ref. [3] for his definition of essential/inessential couplings. But I agree that not all the ambitious goals of the authors might be in reach in this way.
At this point of the refereeing process I believe that the arguments have been exchanged between the authors and me. I have formulated some reservations but that does not mean that the paper should not be published. It is of interest to a wider community of researchers and certainly a useful contribution to ongoing scientific developments and discussions. The authors have also improved the readability and clarity of the text in the revised version. I would therefore like to suggest publication of the manuscript in SciPost in its present form.

---

## Round 3 · Referee Report · Anonymous · 2022-6-7

Report
The authors propose a new scheme within the exact renormalization group formalism which purpose is to follow only the flow of the essential couplings while systematically discarding the inessential ones. This is done by exploiting the invariance under frame transformations to constrain the flow to the subspace spanned by essential couplings while fixing the inessential ones through suitable prescriptions.
While building on previous developments within perturbative and exact RG, this paper provides a systematic way of eliminating {\it all} inessential couplings. The idea is clearly explained and discussed with respect to previous works. An explicit implementation is provided on the example of the 3D Ising model at order 2 of the derivative expansion.
The gain in terms of simplification of the equations to be solved and computational cost is well stressed. However, some comments about the accuracy could be added. How do the numerical results for the critical exponents compare with the standard scheme ? A small table summarizing the results in the standard scheme (both after optimization and for the same regulator choice (144)) and in the essential scheme would be useful.
The whole scheme is definitively appealing and conceptually elegant, even if to fully assess its merits would require to confront it to more involved situations, such as higher orders of the derivative expansion or any of the applications mentioned in Sec. X, in particular to grasp to which extend its practical implementation remains feasible and advantageous. While this is clearly beyond the scope of this paper, it certainly appears as a route worth exploring and it will be interesting to follow what it yields. For this reason and as a conclusion, I recommend the publication of this article in SciPost.

---

## Round 3 · Author Response

We thank the referee for their report and constructive criticisms of our work. We have addressed the main concerns in our new submission. Below we respond to each of the points that have been raised and summarise the corresponding additions to our paper.
(1) In the new version we have clarified our statements. Without the referee giving specific examples, it is hard to know which statements are “not supported enough by factual arguments”.
We are developing a new approach to the exact renormalisation group that includes both practical and conceptual novelties which we believe will have many useful applications. As a result, we have endeavoured to give an extensive explanation of the formalism and conceptual ideas alongside the application to the 3D Ising model, which exemplifies our new approach. Regarding making substantial cuts, we would not be comfortable removing any further text, because this operation would compromise the presentation of the draft, in our opinion. We have nonetheless reflected on the wording of some statements in the text to best remove any ambiguities.
(2) The simple propagator is not the propagator of the physical field but rather that of the parameterised field.
We have stressed this point further throughout the paper. All the physical information can be found by computing the propagator of the physical field denoted by $\hat{\chi}$. We have now added section 3B which explains how observables, such as the two-point function of the physical field, can be computed using the composite operator flow equation (85).
(3) We have been more explicit about the manner in which the theory can be regularised if the UV cutoff is kept finite. Largely, we have worked in the formal limit where the UV cutoff is removed, which is justified when applying the renormalisation group to critical phenomena since the correlation length diverges. We have, however, discussed at length how observables can be computed when a cutoff is present in the new section 3B. For example, the microscopic action in the presence of a UV cutoff is now written down in equation (29) and enters into the formalism as the initial condition for the flow equation.
Since the RG kernel is determined by solving the flow equation, it will be dependent on the form of the UV regularisation.
This is true also in the standard formulation of the effective average action when the UV cutoff is kept finite as has been discussed first by Morris (ref[15]). However, since it is not the main focus of our work to discuss the case where a finite cutoff is kept, we have been brief. For example, we have not discussed how the minimal essential scheme would be modified in the presence of a UV cutoff.
(4) It is true that we have used an implicit construction and we concede that we have been rather brief on this point in the original draft. We have now expanded on this point considerably. Our approach is based on a self-consistent determination of $\Psi_k$ by imposing renormalisation conditions on the form of the effective action to fix the values of the inessential couplings.
The underlying assumption that we make is that there exists a $\hat{\phi}_k[\hat{\chi}]$ such that these constraints on the form of the effective average action can be achieved.
However, the form of $\hat{\phi}_k[\hat{\chi}]$ does not need to be known explicitly within our formalism.
To be concrete, we have established that observables do not depend on the form of $\hat{\phi}_k[\hat{\chi}]$ and that a change in the form of $\hat{\phi}_k[\hat{\chi}]$ is equivalent to the change of an inessential coupling. Moreover, as we have explained in section 3B, we can compute observables despite the lack of knowledge of $\hat{\phi}_k[\hat{\chi}]$.
To be clear, $\Psi_k$ is determined self consistently by solving the flow equation rather than being chosen a priori. Therefore, within a given approximation, $\Psi_k$ is solved for rather than being chosen freely as the referee has suggested.
(5) We achieved a formalism where we can fix the values of inessential couplings along the RG flow.
This can be realised at the level of a modified flow equation since this provides a term proportional to $\Psi_k$ which is precisely a redundant operator conjugate to an inessential coupling.
Although the proposal of the referee could have some utility, the corresponding non-linear change of the field $\phi$ would not produce a term proportional to a redundant operator.
Put differently, by merely making a transformation of the standard flow equation, the flow of the inessential couplings cannot fixed. To be more specific, inessential couplings are related to the field \hat{\phi} not to $\phi$. Indeed $phi$ is actually a reparameterisation of the source $J$ (via the Legendre transform) not the dynamical variables.
The definition of an inessential coupling is always the same and has been introduced previously by Wegner for the Wilsonian effective action and by Weinberg for the 1PI effective action.
We have shown how the redundant operator of the effective average action is related to those of Wegner (ref [2]) and Weinberg (ref[3]) which clarifies that the definition of inessential couplings is the same. We have added section 2H to make the relation to Wegner’s work transparent.
(6) We have now further elaborated on the definition of an inessential coupling at the classical level in section 2A and given a non-linear example to help the reader understand the concept before the more formal construction is developed.
(7) We have modified the captions of the figures to make them clearer.
(8) Can the referee be more specific about what is not clear with this equation? Examples of the dimensionless redundant operator are given in section 5

---

## Round 3 · List of Changes

In order to meet the referee's comments, we substantially changed the manuscript. Here we report a list of the major changes.
- In Section 2A we have defined what is meant by an inessential classical coupling and given some concrete examples;
- In Section 2B we provided an explicit example of UV regularisation. This has led to further appropriate changes throughout the manuscript;
- Throughout the manuscript we have stressed the difference between the physical field and the parameterised field (see for example the comment after Eq.(38) and at the end of Section 5);
- We added Section 2H to clarify the connection between redundant operators of the Wilsonian effective action and those of the effective average action. In particular, the latter is the expectation value of the former (see Eq.(71)).
- At the end of Section 3A, we have elaborated on how the RG kernel is determined via a self-consistent approach by solving the flow equation;
- We added Section 3B in which we explain how observables can be computed;
- We have added to Section 6C a discussion on how the physical field can be defined at a fixed point;

You are currently on this page

---

## Editorial Decision

editorial_decision: